# Lead Exposure Can Affect Early Childhood Development and Could Be Aggravated by Stunted Growth: Perspectives from Mexico

**DOI:** 10.3390/ijerph20065174

**Published:** 2023-03-15

**Authors:** Leonel Córdoba-Gamboa, Ruth Argelia Vázquez-Salas, Martin Romero-Martínez, Alejandra Cantoral, Horacio Riojas-Rodríguez, Sergio Bautista-Arredondo, Luis F. Bautista-Arredondo, Filipa de Castro, Marcela Tamayo-Ortiz, Martha María Téllez-Rojo

**Affiliations:** 1Dirección de Salud Ambiental, Centro de Investigación en Salud Poblacional, Instituto Nacional de Salud Pública, Cuernavaca 62100, Morelos, Mexico; roberto.cordoba@insp.edu.mx (L.C.-G.);; 2Dirección de Salud Reproductiva, Centro de Investigación en Salud Poblacional, Instituto Nacional de Salud Pública, Ciudad de México 14080, Morelos, Mexico; 3Centro de Investigación en Evaluación y Encuestas, Instituto Nacional de Salud Pública, Cuernavaca 62100, Morelos, Mexico; 4Departamento de Salud, Universidad Iberoamericana, Ciudad de Mexico 01219, Morelos, Mexico; 5Centro de Investigación en Sistemas de Salud, Instituto Nacional de Salud Pública, Cuernavaca 62100, Morelos, Mexico; 6Centro de Investigación en Nutrición y Salud, Instituto Nacional de Salud Pública, Cuernavaca 62100, Morelos, Mexico; 7Research, Evidence, and Learning, Department of Education and Child Population, Save the Children, 501 Kings Highway East, Suite 400, Fairfield, CT 06825, USA; 8Unidad de Investigación de Salud en el Trabajo, Instituto Mexicano del Seguro Social, Ciudad de México 6720, Morelos, Mexico

**Keywords:** early childhood development, language, lead exposure, stunted growth

## Abstract

Background: Lead can affect early childhood development (ECD) differentially due to nutritional deficiencies that lead to stunted growth, defined as being at least two standard deviations below the average height-for-age. These deficiencies are more frequent among children living in rural locations or with lower socioeconomic status (SES); however, studies at a population level are scarce worldwide. Early childhood development plays a crucial role in influencing a child’s health and wellbeing throughout life. Therefore, the aim of this study was to analyze how stunted growth can modify the association between lead exposure and ECD in children from disadvantaged communities. Methods: Data were analyzed from the 2018 National Health and Nutrition Survey in localities with fewer than 100,000 inhabitants in Mexico (ENSANUT-100K). Capillary blood lead (BPb) levels were measured using a LeadCare II device and dichotomized as detectable (cutoff point ≥ 3.3 µg/dL) and non-detectable. As a measure of ECD, language development was assessed in *n* = 1394 children, representing 2,415,000 children aged 12–59 months. To assess the association between lead exposure and language z-scores, a linear model was generated adjusted by age, sex, stunted growth, maternal education, socioeconomic status, area, region (north, center, south), and family care characteristics; afterwards, the model was stratified by stunted growth. Results: Fifty percent of children had detectable BPb and 15.3% had stunted growth. BPb showed a marginal inverse association with language z-scores (β: −0.08, 95% CI: −0.53, 0.36). Children with detectable BPb and stunted growth had significantly lower language z-scores (β: −0.40, 95% CI: −0.71, −0.10) than those without stunted growth (β: −0.15, 95% CI: −0.36, 0.06). Conclusions: Children with stunted growth are more vulnerable to the adverse effects of lead exposure. These results add to previous research calling for action to reduce lead exposure, particularly in children with chronic undernutrition.

## 1. Introduction

Children living in poverty, in rural areas, or with malnourishment are more vulnerable to environmental exposure to lead [1] due to socioeconomical disadvantages, household conditions [2], and living outside major cities or in rural areas near lead–acid battery recycling and smelting sites [2,3,4]. They also have lower overall health and nutritional outcomes and a higher prevalence of poor physical health, including stunted growth [1,3,5,6]. Furthermore, their physical, social, emotional, and cognitive development can be severely affected and translate to lower opportunities later in life [1,5,6]. Moreover, the burden of lead (Pb) exposure is higher in low- and middle-income countries [7]. Global estimations show that children with lower socioeconomic status (SES) are more prone to Pb exposure [8] and also have higher rates of undernutrition [9]. Children who are undernourished are more vulnerable to Pb exposure [10] since the inadequate intake of micronutrients favors Pb absorption, increasing its toxicity [11]. Therefore, early-life exposure to lead can directly impact early childhood development (ECD) in children with undernutrition.

Early childhood is the period that comprises the first eight years of life; during this period, the brain’s plasticity defines a child’s characteristics, and the development of motor and language skills, thinking, feelings, learning, and social skills are enhanced [1]. During pregnancy, human brain development involves the synaptogenesis formation of higher cognitive and receptive language functions. Brain cells are made, axons and dendrites grow, and there is development in the synapses for the supportive tissues that surround the nerve cells. Once the nerve cells are formed, they rapidly extend axons and dendrites and begin to form connections with each other [1]. These processes increase synapsis production until four years of age. At eight years of age, the higher cognitive functions start to decrease through adolescence and beyond [1]. Children’s language skills are a core component of ECD, promoting thinking, problem solving, and the development of interpersonal relationships [1]. Language is considered a good approximation of overall ECD, with an acceptable predictive validity of children’s global intellectual coefficients [12,13]. Other studies have shown that language strongly predicts future academic achievement and socio-emotional adaptation [14,15]. Synaptogenesis in the parts of the brain responsible for receptive language and speech production begins during the first months of life and reaches its peak at around eight to nine months [1]. Previous studies have linked early life lead exposure to language development [16] and a reduction in the activation of the brain’s left frontal cortex, adjacent to Broca’s area and Wernicke’s area, both related to language and learning [17]. In early life, every 10 μg/dL increase in blood Pb (BPb) results in a loss of five points on language tests [16]. 

Several studies in Mexico have documented Pb neurobehavioral toxicity [18,19,20,21,22,23,24,25,26,27,28], including its effect on language [29,30]. However, they all used non-representative samples with narrow geographical dispersion, such as around a specific exposure source (e.g., a mining site). In 2018, the National Health and Nutrition Survey of localities with fewer than 100,000 inhabitants (ENSANUT-100K) documented that more than one million children had BPb above 5 µg/dL (the reference value according to the official Mexican standard) [31]. This national survey also documented a 15% prevalence of stunted growth in children aged 0–59 months [32]; the localities assessed in the survey have the highest prevalence of poverty in Mexico and are also home to 57% of children aged 0–59 months [33]. 

Blood lead reference values are currently under review in Mexico due to the strong evidence that children’s health can be affected by BPb levels as small as 3.5 µg/dL [34]. A new review by the Centers for Disease Control and Prevention of the United States (CDC) decreased the blood lead reference value to 3.5 µg/dL [35]. Therefore, the objective of this study was to analyze the association between lead exposure and ECD in a representative sample of children aged 12–59 months living in localities with fewer than 100,000 inhabitants in Mexico and to assess an effect modification of this association by chronic undernutrition. The hypothesis of this study was a negative association between lead exposure and ECD, with a stronger negative association in children with stunted growth. 

## 2. Methods

The ENSANUT-100K was a cross-sectional study, conducted between March and June 2018, that employed a household probabilistic survey using basic geostatistical areas (AGEB, acronym in Spanish) as the primary sampling unit [30]. The study provided a national and regional representative survey of the population in localities with 100,000 inhabitants or fewer and was designed to over-represent disadvantaged households. Thus, ENSANUT-100K allocated higher probabilities of selection (higher sampling weights) to disadvantaged households; therefore, all estimations of this paper are weighted in order to take into account the probabilities of selection. 

The survey’s regions were north (eight states), center (twelve states), and south (eleven states). Depending on the child’s age, the ENSANUT-100K collected data on different child health and wellbeing indicators, including nutritional status (blood samples for the following micronutrients: iron, zinc, vitamin A, vitamin B12, and vitamin D [36]), lead exposure, and ECD determinants and status. ENSANUT-100k evaluated children under 60 months because evidence shows that carrying out actions aimed at children in this age range can have greater positive impacts and provide benefits to their present development and throughout their lives [37,38,39]. For all children under 60 months (*n* = 3576), the survey collected information about ECD and some ECD determinants using the ECD questionnaire. Language development data were collected for all children between 12 and 59 months of age (*n* = 2931). Children between 36 and 59 months (*n* = 1553) were further eligible for collecting data on the Early Childhood Development Index (ECDI) (Figure 1). One child under 60 months old was randomly selected to measure capillary blood lead levels (BPb) and anthropometric measurements (*n* = 1594) in each household. Thus, the study population included *n* = 1394 children aged 12–59 months who had complete interviews, including a language development assessment, a blood lead measurement, and anthropometric measurements. Of these, *n* = 753 children aged 36–59 months had complete ECD-status data including both language development and the ECDI (Figure 1). 

### 2.1. Language Assessment

Language development was measured using two instruments according to the children’s ages; for 12–42 month olds, the short Spanish versions of the MacArthur–Bates Communication Development Inventories I, II, and III (CDI-I, II, and III) were administered to the child’s mother. For 43–59 month old children, the Spanish version of the Peabody Picture Vocabulary Test (PPVT-III) was administered directly to the child. Both instruments have previously been validated in Spanish, and they are considered suitable for use in the context of household surveys [40]. Both instruments have construct validity (CDI of 0.74–0.93 and PPVT-III 0.84) and adequate reliability (CDI from 0.97 to 0.99 and PPVT-III 0.91) [41,42]. Interviewers were trained and standardized before field work, with an agreement level greater than 80% in Cohen’s Kappa test. Bland–Altman graphs were used to identify overestimation or underestimation scoring biases among the interviewers [40,43]. In indigenous communities, when necessary, there was a translator [40]. For children aged 12–42 months, the language raw score was measured as the number of words that each child said in CDI-I, II, and III; for children aged 43–59 months, language raw score was based on the PPVT-III by subtracting the total number of errors from the maximum number of correct words reached by each child. Language scores were standardized by generating a single variable with a mean of zero and a standard deviation of one (language z-score) using a local polynomial regression, which considers the survey weights and adjusts by the age of children in months [40]. 

### 2.2. Early Childhood Development Index (ECDI)

The ECDI was developed and validated by the United Nations Children’s Fund (UNICEF) and comprises 10 questions which are administered to the mother to learn about the child’s performance and behaviors (yes vs. no). The ECDI is used to assess 36–59 month old children and considers four domains: literacy and numerical knowledge (three questions), physical development (two questions), socio-emotional development (three questions), and learning (two questions) [44,45]. Children are considered “on-track” for those domains with two questions that have at least one positive answer, and for those with three questions that have at least two positive answers. When at least three out of the four domains are on-track, the ECDI identifies a child as being “developmentally on-track” (i.e., <3 domains = not on-track vs. ≥3 domains = on-track) [45].

### 2.3. Blood Lead Measurements 

BPb concentrations were analyzed through voltammetry using capillary blood with a LeadCare II portable device (Magellan Diagnostics, North Billerica, MA, USA, sensitivity (88%) and specificity (99%) [46]); the detection limit ranges from 3.3 to 65 μg/dL. A blood sample of 50 μL was obtained by capillary puncture with a minimally invasive procedure. The method is validated and approved by the United States Center for Disease Control and Prevention (CDC). Trained personnel drew the samples following a standardized protocol that included children washing their hands with soap and water and not touching anything prior to collecting the sample [47]. Nevertheless, despite this procedure, lead residues might have been present on the children’s fingers [47]. In the ENSANUT-100K, the BPb measurements were conducted with the LeadCare II—instead of the gold standard venous blood lead measurement—due to limited financial resources and the LeadCare II’s portability and suitability in a nationwide home-based survey [48]. For these analyses, BPb concentrations were used as a dichotomous variable: non-detectable BPb (BPb < 3.3 μg/dL) vs. detectable BPb (BPb ≥ 3.3 μg/dL). Additionally, to evaluate the association among different levels of lead exposure, a categorical BPb variable using three categories was used: non-detectable BPb (BPb < 3.3 μg/dL), detectable from 3.3 μg/dL to 5 μg/dL, and detectable above 5 (BPb > 5 μg/dL) (5 µg/dL is the reference value currently under review in Mexican health guidelines). 

### 2.4. Anthropometric Measurements

Height and length were measured by trained personnel, following standard protocols [32]. For 12–24 month old children, length was measured with the child lying down using an infantometer with a precision of 1 mm. Height in 25–59 month old children was measured with the child standing, without shoes, using a stadiometer with a precision of 2 mm [32]. Length/height-for-age was calculated and transformed into z-scores using the World Health Organization (WHO) reference, and length/height-for-age z-scores between −6.0 and +6.0 with respect to the median of the reference population were considered valid [49]. Children with a length/height-for-age z-score at least two standard deviations from the referenced norm were classified with stunted growth.

### 2.5. Covariates

Information was included on the child’s sex (girl vs. boy) and age (months), as well as on family care characteristics such as the child’s attendance at daycare (12–35 months) or preschool (36–59 months) (no vs. yes), learning support at home (no vs. yes), children’s access to books (<3 vs. ≥3 books), availability of types of toys (<2 vs. ≥2 toys types), and exposure to violent discipline methods (no vs. yes). The family care characteristics are standardized indicators of UNICEF’s Multiple Indicator Cluster Surveys (MICS); as such, we followed the standardized methodology of MICS, allowing us an international comparison of these indicators [45]. 

Furthermore, information was included on maternal educational level (elementary school vs. middle school or higher) as well as household SES (lower, middle, higher), locality type (rural vs. urban), and region (north, center, south). SES was constructed as an index based on a principal component’s analysis of the variables that describe housing conditions and assets [33]. Further details about this analysis can be found in the ENSANUT-2018 description [33].

### 2.6. Statistical Analysis

The mean and standard errors of language z-scores were estimated; also, the prevalence and 95% confidence intervals of children, maternal, and family care characteristics by stunted growth were estimated and compared using t-tests and chi-square tests, respectively. Confidence intervals that excluded zero were used to identify parameters statistically different from 0, and *p*-values of <0.05 were used to identify parameters statistically different from 0. The normality of language z-score was evaluated through the Shapiro–Wilk test. To evaluate the association between BPb and language z-score, a linear regression model was used which was stratified by stunted growth to assess a potential differential effect. Models were adjusted by age, sex, learning support, having books at home, type of toys, exposure to violent discipline, maternal education, locality type, country region, and SES. All potential confounders were selected based on the literature review [16,34,40]. To test the interaction of lead exposure and stunted growth, another model was performed including this interaction term in a regression model where z-score was the response variable. As a post hoc test, a goodness of fit analysis and a standard analysis of the residuals were conducted. To analyze the association between BPb and “Not on-track” ECDI, a multivariable logistic regression model, adjusted by the same set of confounders and stratified by stunted growth under the same rationale as before, was conducted. Additionally, the interaction term of lead exposure and stunted growth was performed. All statistical analyses considered the complex survey design [50] using Stata 15.0 software’s svy command (College Station, TX).

## 3. Results

Table 1 shows the characteristics of the sample that included *n* = 1394 children, representing 2,415,000 12–59 month old children. Just over half of the children were between 12 and 35 months (50.7%) and were girls (52.7%, 95% CI: 44.9%, 60.4%). Most of the mothers had middle school or higher education (76.7%, 95% CI: 69.8%, 82.5%); 62.1% of households were classified as low-SES (95% CI: 52.0%, 71.2%); 54.4% of households were in rural communities (95% CI: 42.5%, 65.7%); and 52.3% of children lived in the south region of the country (95% CI: 41.6%, 62.7%), where stunted growth prevalence was higher (65.5%, 95% CI: 49.3%, 78.8%), followed by the central region (18.7%, 95% CI: 10.5%, 31.2%) and the north region (15.8%: 95% CI: 6.9%, 32.1%). According to family care characteristics, the percentage of children with learning support was 75.5% (95% CI: 68.7%, 81.3%).

The mean language z-score was −0.08 (SE: 0.07) and the prevalence of stunted growth was 15.3%. Compared to children without stunted growth (−0.09; SE: 0.07), children with stunted growth had lower language z-scores (−0.02; SE: 0.15; *p* = 0.67) (Table 1). Language z-scores were lowest in children from the south region (−0.17, SE: 0.09), followed by the central (−0.06, SE: 0.11) and north regions (0.18, SE: 0.15) (*p* < 0.001) (data not shown in tables). The prevalence of children with detectable BPb was 50.0% (95% CI: 39.2%, 60.7%) (Table 1), and the prevalence of children with levels above 5 µg/dL was 23.3% (95% CI: 11.8%, 40.8%). Meanwhile 56.6% of the children in the south region (95% CI: 38.6%, 73.0%), 45.0% in the central (95% CI: 36.4%, 53.9%), and 36.4% in the north (95% CI: 26.5 %, 47.5%) had detectable BPb (data not shown in tables).

Table 2 shows the characteristics of the subsample of children aged 36 to 59 months, of whom 24.4% were not developmentally on-track (95% CI: 18.2%, 31.9%). A total of 45.0% had detectable blood lead levels (95% CI: 36.9%, 53.3%), and 16.7% had stunted growth (95% CI: 10.4%, 25.6%). In this age group, most of the children were between 36 and 47 months old (57.4%, 95% CI: 49.3%, 65.2%) and around half of them were girls (52.1%, 95% CI: 42.6%, 61.4%). Most of their mothers had middle school or higher education (75.0%, 95% CI: 66.6%, 81.8%), and most of the households reported having a lower SES (53.2%, 95% CI: 43.1%, 63.0%) and lived in rural communities (55.7%, 95% CI: 45.8%, 65.2%). Half of the children in this age group lived in the south region (49.8%, 95% CI: 40.6%, 58.9%). Family care characteristics showed that most of the children had more than two types of toys (99.4%, 95% CI: 98.6%, 99.7%), and the prevalence of children disciplined with violent methods was 59.9% (95% CI: 50.8%, 68.4%) (Table 2). According to the ECDI, 27.4% of children with detectable BPb were not developmentally on-track (95% CI: 17.1%, 40.7%) compared to 22.2% of children with non-detectable BPb (95% CI: 15.1%, 31.2%) (*p* = 0.47). One quarter of children not developmentally on-track had stunted growth (25.3%, 95% CI: 10.9%, 48.4%). The percentage of children not developmentally on-track was significantly higher among children living in the poorest households (*p* < 0.001) (Table 2).

A negative but not statistically significant association between detectable BPb and language z-score was observed (β = −0.08, 95% CI: −0.53, 0.36; *p* = 0.711) (Table 3). Although the coefficients for the covariates were in the expected direction, the only one that showed a significant relationship (*p* < 0.05) was having fewer than three books (β = −0.30, 95% CI: −0.58, −0.02; *p* < 0.03). In the stratified analysis, a significant and negative association between detectable BPb and language z-score was identified among the group of children with stunted growth (β = −0.40, 95% CI: −0.71, −0.10; *p* < 0.01) (Table 3). The association between BPb and language z-score among non-stunted-growth children was not statistically significant (β = −0.15, 95% CI: −0.36, 0.06; *p* = 0.158). Other characteristics such as sex, locality type, learning support, and access to children’s books differed with language z-scores depending on stunted growth. 

Table 4 shows the association between BPb and not being developmentally on-track according to the ECDI. An OR of 1.09 for children with detectable BPb compared to non-detectable BPb was observed (95% CI: 0.57, 2.07; *p* = 0.783). Other associated risk factors were identified as the SES (middle OR = 4.28 (95% CI: 2.27, 8.05; *p* < 0.001); higher OR = 12.7 (95% IC: 6.21, 25.7; *p* < 0.001)) and the region (center OR = 2.73 (95% CI: 1.09, 6.80, *p* < 0.03); south OR =3.24 (95%IC: 1.32, 7.94; *p* < 0.01)). Children that were not exposed to violent discipline methods had an OR = 0.13 (95%IC: 0.06, 0.28; *p* < 0.001) (Table 4). Stratified analysis by stunted growth did not show any statistically significant differences (data not shown in tables). 

Results for the three-category BPb are shown in the Appendix A. For the language analyses, a null association was observed in the overall analysis; however, in the stratified analyses a negative association was found (β = −0.67, 95% CI: −1.02, −0.32) among stunted-growth children with 3.3–5 µg/dL BPb. A null association was also observed for non-stunted-growth children (Appendix A). For the ECDI analyses, an OR of 2.32 was observed (95% CI: 0.99, 5.40) for not being developmentally on-track among those children with BPb > 5 µg/dL, and an OR of 0.57 (95% CI: 0.28, 1.14) was seen for children with BPb between 3.3 and 5 µg/dL compared to children with non-detectable BPb (Appendix A).

## 4. Discussion 

To our knowledge, this is the first study to assess the association between lead exposure and ECD in a representative sample of children in Mexico, and it is one of the few studies that have evaluated this association in children with chronic undernutrition [51]. The results of this study confirm that lead exposure affects language development in vulnerable populations (stunted-growth children). Among all children, language z-scores were lower in children with detectable BPb (β = −0.08; 95% CI: −0.53, 0.36). This association was modified by stunted growth; children with chronic undernutrition and detectable lead exposure were more likely to have a lower language z-score than children without stunted growth (β = −0.40; 95% CI: −0.71, −0.10). In the analysis of three levels of lead exposure, a similar pattern was observed; the analysis stratified by stunted growth showed a lower language z-score among children with 3.3–5 µg/dL BPb, and a null association for the category above 5 µg/dL BPb (Appendix A). The results from this study indicate that simultaneous exposure to lead and chronic undernutrition are vital characteristics to consider when studying child development, especially cognitive deficits [51,52]. The interaction between lead exposure and stunted growth was tested; however, due to the variability in child characteristics in the entire sample, the interaction term was not statistically significant (*p* = 0.820). However, through the stratified analysis, the effects of lead exposure in stunted-growth children were observed—possibly due to the minimization of variability in children in this stratum—supporting the effect modification of stunted growth (Table 3).

Children start developing language from their first months of age, when brain synapse function normally begins [1]. Several studies have documented that lead exposure affects children’s language development by showing an association between BPb and brain regions related to language development [16,30]. Additionally, studies have shown that lead exposure affects children’s heights due to the relation between BPb and abnormalities in growth hormones, and BPb’s interaction with essential micronutrients [51,53,54,55,56]. Language development can be affected in stunted children due to their compromised nutritional status. They may absorb more metals or other toxicants that affect language and inhibit neurodevelopment [51]. Even though we know that deficiencies in some micronutrients, such as iron, can increase children’s vulnerability to lead exposure [11], the prevalence of anemia in this sample (<100.0 Hb g/L) was 16.7% and we did not find differences in lead exposure or language development (data not shown). 

Gleason and colleagues found that stunted growth modified the adverse effect of lead, showing that stunted-growth children with BPb levels similar to those in this study had more substantial cognitive effects [51]. A prospective study among pregnant women suggested that early childhood may be a critical window for lead effects on stunted-growth children, as those with nutritional deficiencies may absorb more lead, leading to greater alterations in childhood development [51,57]. 

Stunted growth results from chronic or recurrent malnutrition and is potentially influenced by poverty, SES, healthcare access, and maternal malnutrition. Children with stunted growth can suffer irreversible physical and cognitive damage that can last a lifetime [9]. Several countries have seen a decline in the prevalence of stunted growth, but countries in Central America, Africa, and Asia maintain a prevalence of above 30% [9]. Likewise, some of these countries have a higher prevalence of lead exposure [7], making these results relevant for different populations with the same vulnerable conditions. 

The results of this study are consistent with other studies that found that lead exposure above 3.3 μg/dL is associated with a sub-optimal development of language skills [16,17,30]. Research in the United States found that every 10 μg/dL increase in blood BPb results in a loss of five points on language tests, and research in Mexico City found significant negative associations between language score and the presence of lead in concentrations from 2 to 5 μg/dL [16,30]. Other studies have found significant associations between increases in BPb (every 5 μg/dL) and brain activation in regions associated with language function and development, even when overall cognitive measures showed non-significant associations [17,58]. These results indicate that lead exposure may affect language development even at what has been considered low lead levels (<2 to 10 µg/dL). 

An increased correlation was observed between detectable BPb levels and children not being developmentally on-track, although the results were not statistically significant (OR = 1.09, 95% CI: 0.57, 2.07). However, when we analyzed three levels of lead exposure, we found an increased chance of not being developmentally on-track in children who had BPb > 5 µg/dL, independent of stunted growth condition (OR = 2.32 95% CI: 0.99, 5.40) (Appendix A). The discrepancy in these results could be due to the small sample size, but this association is plausible given the knowledge of the effects of lead exposure on children’s development. Additionally, not being developmentally on-track according to the ECDI has been associated with environmental exposures [59] and with other factors such as low SES, lack of early stimulation, and malnutrition [60,61,62,63]. 

An association between stunted growth and not being developmentally on-track was observed but was not statistically significant (data not shown in tables), probably due to the small sample size since only 42 children were identified as not being developmentally on-track and had stunted growth. However, the prevalence of stunted growth was similar in the complete sample (15.3%) compared to those children with an ECDI evaluation (16.7%). 

In Mexico, several studies have explored the relationship between lead exposure and ECD [19,23,28,30,53]. Nonetheless, in these studies, language was studied as a component of a battery of tests measuring overall ECD that are difficult to administer to a survey population [30]. In contrast, the ECDI is a standard measure in population-based surveys; results from the ECDI in Mexico in 2018 showed a prevalence of 19.9% of children who were not developmentally on-track, with an increasing possibility of being on-track in children with higher SES and among those with ownership of children’s books [64]. 

Family care characteristics are important to consider in ECD. Children with no learning support and with fewer than three books had a diminishing language z-score, and children who were not disciplined with violent methods had better odds of being developmentally on-track. An adequate family environment, with early stimulation and without violent discipline methods, could improve ECD.

Characteristics by region are also essential to consider. The south region of the country had a higher prevalence of detectable BPb, a higher prevalence of stunted growth among children, and lower language z-score results compared with the north region. These differences by region suggest that children in the south region are more vulnerable to the effects of lead exposure than children in the north region; however, the interaction between lead exposure and region was tested but was not statistically significant (center *p* = 0.905, south *p* = 0.548). The results by region are consistent with other studies which have found that the south region has a higher prevalence of detectable BPb and a greater use of lead-glazed pottery [34]. Moreover, this region has the poorest socioeconomic and family care conditions [40], and children in the south region also have the lowest language z-scores [40]. The south region has the highest prevalence of stunted growth in Mexico, and although this prevalence has been slowly decreasing through the years, the differences in stunted growth rates between regions are increasing, especially between the north and south regions [65]. The south region has the lowest gross domestic product in Mexico compared to the country’s other regions [66]. Although an association by region was not observed, public health actions need to be directed to the states of the south region. 

This research has some limitations. The cross-sectional survey design does not allow us to determine the temporality and the history of lead exposure, the effect on language, and the possible relationship between BPb and stunted growth in this sample. Another limitation is the use of capillary blood tests using LeadCare instead of venous blood samples. LeadCare is not the gold standard for measuring blood lead levels for association studies due to its detection limit and its use of capillary blood (3.3 µg/dL) [48]; however, to address this limitation, statistical analyses were performed by dichotomizing BPb (detectable vs. non-detectable) to group concentrations above the detection limit. Due to the portability, specificity (88%), and sensitivity (99%) [46], LeadCare is a suitable screening method especially in a nationwide survey and for concentrations around 5 μg/dL [67]. Carrying out such a study using venous blood would be extremely logistically difficult and expensive.

Despite these limitations, the study’s strengths include its sample size and national representativeness, which allow us to give a general overview of the relationship between lead exposure, stunted growth, and ECD in Mexico. Another strength of this study is that the results found are biologically plausible and supported by other studies that have found increased BPb associated with ECD deficiencies in cross-sectional studies [68,69]. This study demonstrates that children in vulnerable conditions are exposed to both repeated malnutrition, resulting in stunted growth, and lead exposure, which negatively impacts ECD even at lower levels than those allowed by Mexican authorities (5 µg/dL). As there is no level of exposure to lead that is known to be without harmful effects [7], these results show that with an increase in BPb there is a decrease of −0.40 in language z-score. Additionally, these results contribute to the discussion on reducing permissible blood lead levels worldwide and in Mexico, which seeks to reduce the acceptable limit to 1 µg/dL.

In conclusion, we found lower language z-scores in children with stunted growth and detectable BPb. These results show the need for public health actions directed toward reducing lead exposure, along with dietary interventions in children that will help support their general development. 

## 5. Conclusions

A lower language z-score in children with stunted growth and detectable BPb were found. These results show the need for public health actions directed to reduce lead exposure and implement dietary interventions in children that will help the general development of children.

## Figures and Tables

**Figure 1 ijerph-20-05174-f001:**
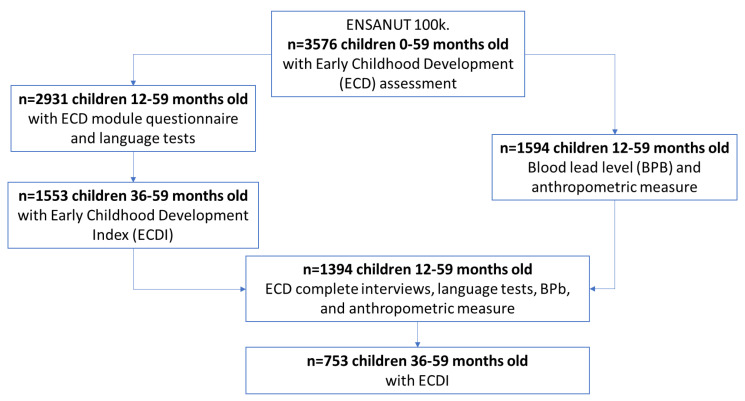
Descriptive diagram of the study population.

**Table 1 ijerph-20-05174-t001:** Child, maternal, household, and family care characteristics by stunting condition.

	Total *n* Sample = 1394 *n* = 2,415,000 (%)	95% CI	Stunting Condition^c^				*p*-Value
			Children with Stunting		Children without Stunting		
			*n* Sample = 256 N = 437,500 (15.3%)	95% CI	*n* Sample = 1138 *n* = 1,945,000 (84.7%)	95% CI	
Child characteristics							
Mean language z-score (SE)	−0.08 (0.07)	−0.15, −0.05	−0.02 (0.15)	−0.32, 0.28	−0.09 (0.07)	−0.24, 0.05	0.67 ^d^
Blood lead level							
Non-detectable (< 3.3 μg/dL)	806 (50.0)	39.3, 60.8	162 (48.7)	32.2, 65.4	644 (50.3)	38.1, 62.4	0.88 ^e^
Detectable (≥ 3.3 μg/dL)	588 (50.0)	39.2, 60.7	94 (52.3)	34.5, 67.7	494 (49.7)	37.6, 61.9	
Age (months)							
12–23	289 (28.7)	18.7, 41.3	43 (23.8)	12.9, 39.6	246 (29.6)	18.3, 41.3	0.78 ^e^
24–35	352 (22.0)	16.6, 28.5	53 (22.5)	11.8, 38.6	299 (21.9)	16.6, 28.5	
36–47	377 (28.3)	22.4, 35.0	87 (26.8)	16.4, 40.6	290 (28.6)	22.4, 35.0	
48–59	376 (20.0)	15.7, 27.4	73 (26.9)	13.4, 46.6	303 (19.9)	15.7, 27.4	
Sex							
Girl	669 (52.7)	44.9, 60.4	126 (42,5)	26.8, 59.9	543 (54.5)	46.1, 62.6	0.22 ^e^
Boy	725 (47.3)	39.6, 55.1	130 (57.5)	40.1, 73.1	595 (45.5)	37.4, 53.8	
Maternal characteristics							
Maternal schooling (%)^a^							
Middle school or higher	876 (76.7)	69.8, 82.5	127 (64.5)	47.1, 78.8	749 (78.9)	71.9, 84.5	0.06 ^e^
Elementary school	436 (23.3)	17.5, 30.2	114 (35.5)	21.2, 52.9	322 (20.1)	15.4, 28.1	
Household characteristics							
**SES (%) ^b^**							
** Lower**	**954 (62.1)**	**52.0, 71.2**	**209 (60.8)**	**45.0, 85.1**	**745 (60.9)**	**46.5, 71.1**	**0.05 ^e^**
** Middle**	**342 (27.3)**	**20.2, 35.9**	**37 (11.6)**	**6.7, 19.4**	**305 (30.2)**	**21.5, 40.5**	
** Higher**	**92 (10.6)**	**6.5, 16.7**	**8 (19.6)**	**6.4, 46.5**	**84 (8.9)**	**6.0, 13.0**	
Locality type							
Urban	218 (45.6)	34.3, 57.5	21 (26.9)	12.5, 48.7	197 (49.0)	36.4, 61.8	0.08 ^e^
Rural	1176 (54.4)	42.5, 65.7	235 (73.1)	36.4, 61.8	941 (51.0)	38.2, 63.6	
Country region							
North	238 (15.4)	10.5, 22.0	19 (15.8)	6.9, 32.1	219 (15.3)	10.3, 22.1	0.13 ^e^
Center	427 (32.3)	25.0, 40.6	51 (18.7)	10.5, 31.2	376 (34.8)	25.9, 44.9	
South	702 (52.3)	41.6, 62.7	182 (65.5)	49.3, 78.8	520 (49.9)	37.2, 62.5	
Family care characteristics							
Preschool education							
Yes	426 (21.8)	18.4, 25.5	89 (17.1)	11.0, 25.5	337 (22.6)	18.9, 26.9	0.21 ^e^
No	968 (78.2)	74.5, 81.5	167 (82.9)	74.5, 88.0	801 (77.4)	73.1, 81.1	
**Percentage of children with learning support (%)**							
** Yes**	**907 (75.5)**	**68.7, 81.3**	**143 (59.6)**	**42.0, 75.0**	**764 (78.4)**	**71.5, 84.0**	**0.02 ^e^**
** No**	**487 (24.5)**	**18.7, 31.3**	**113 (40.4)**	**24.0, 58.0**	**374 (21.6)**	**15.0, 28.5**	
Percentage of children with at least three books (%)							
≥3	235 (21.1)	15.5, 28.0	18 (22.4)	8.6, 46.9	217 (20.9)	15.1, 28.1	0.88 ^e^
<3	1159 (78.9)	71.0, 84.5	238 (77.6)	53.1, 91.4	921 (79.1)	71.9, 84.9	
Percentage of children with two or more toy types (%)							
≥2	1348 (96.6)	93.5, 98.2	243 (97.8)	95.4, 99.0	1105 (96.4)	92.7, 98.2	0.30 ^e^
<2	46 (3.4)	1.7, 6.5	13 (2.1)	1.0, 4.6	33 (3.6)	1.8, 7.3	
Children disciplined with violent methods (%)							
Yes	774 (58.0)	48.4, 67.1	129 (57.3)	40.2, 72.7	645 (58.2)	47.4, 68.2	0.92 ^e^
No	620 (42.0)	32.9, 51.5	127 (42.7)	27.2, 59.8	493 (41.8)	31.8, 52.5	

^a^ Missing value = 83; ^b^ Missing value = 6; ^c^ Stunting condition: no stunting (>−2 standard deviation height-for-age); stunting (≤−2 standard deviation height-for-age); ^d^ T-test ^e^ Chi-square.

**Table 2 ijerph-20-05174-t002:** Early Childhood Development Index (ECDI) by children, mother, household, and family care characteristics.

	Total *n* Sample = 753 *n* = 1,300,000 (%)	95% CI	On-Track *n* Sample = 569 *n* = 982,000 (75.6%)	95% CI	Not On-Track *n* Sample = 184 *n* = 318,000 (24.4%)	95% CI	*p*-Value ^d^
Child characteristics							
Blood lead level							
Non-detectable (< 3.3 μg/dL)	426 (55.0)	46.6, 63.0	321 (77.8)	68.8, 84.7	105 (22.2)	15.1, 31.2	0.47
Detectable (≥ 3.3 ug/dL)	327 (45.0)	36.9, 53.3	248 (72.6)	59.2, 82.8	79 (27.4)	17.1, 40.7	
Stunting condition^a^							
No stunting	593 (83.3)	74.4, 89.6	451 (75.6)	67.2, 82.4	142 (24.4)	17.6, 32.8	0.93
Stunting	160 (16.7)	10.4, 25.6	118 (74.7)	51.6, 89.1	42 (25.3)	10.9, 48.4	
Age (months)							
36–47	377 (57.4)	49.3, 65.2	262 (71.1)	58.6, 81.0	115 (28.9)	18.9, 41.3	0.12
48–59	376 (42.6)	34.8, 50.7	307 (81.3)	73.6, 87.1	69 (18.7)	12.9, 26.4	
Sex							
Girl	357 (52.1)	42.6, 61.4	279 (77.1)	67.1, 84.7	78 (22.9)	15.3, 33.0	0.64
Boy	396 (47.9)	38.6, 57.4	290 (73.6)	60.8, 83.4	106 (26.3)	16.3, 38.5	
Maternal characteristics							
Maternal schooling (%)^b^							
Middle school or higher	449 (75.0)	66.6, 81.8	347 (74.9)	65.1, 82.8	102 (25.1)	17.2, 34.9	0.87
Elementary school	249 (25.0)	18.2, 33.4	181 (73.7)	58.3, 84.8	68 (26.3)	15.1, 41.7	
Household characteristics							
**SES (%)^c^**							
** Lower**	**511 (53.2)**	**43.1, 63.0**	**373 (65.5)**	**54.9, 74.7**	**138 (34.5)**	**25.3, 45.1**	**<0.001**
** Middle**	**183 (33.7)**	**23.7, 45.5**	**145 (85.6)**	**76.8, 91.4**	**38 (14.4)**	**8.6, 23.2**	
** Higher**	**53 (13.1)**	**7.2, 22.7**	**47 (91.2)**	**88.6, 93.1**	**6 (8.8)**	**6.8, 11.4**	
Locality type							
Urban	112 (44.3)	34.7, 54.2	87 (76.6)	61.7, 86.9	25 (23.4)	13.0, 38.3	0.77
Rural	641 (55.7)	45.8, 65.2	482 (74.5)	66.7, 80.9	159 (25.5)	19.0, 33.3	
Country region							
North	124 (17.1)	10.8, 25.9	100 (85.9)	76.0, 92.1	24 (14.1)	7.8, 23.9	0.06
Center	234 (33.1)	26.3, 40.5	181 (79.2)	70.5, 85.8	53 (20.8)	14.2, 29.4	
South	379 (49.8)	40.6, 58.9	273 (69.1)	55.3, 80.1	106 (30.9)	19.9, 44.6	
Family care characteristics							
**Preschool education**							
Yes	382 (39.4)	31.7, 47.6	311 (75.0)	60.6, 85.4	71 (25.0)	14.6, 39.4	0.92
No	371 (60.6)	52.4, 68.3	258 (75.7)	67.1, 82.6	113 (24.3)	17.3, 32.8	
Percentage of children with learning support (%)							
Yes	473 (75.5)	68.4, 81.4	377 (77.1)	67.8, 84.4	96 (22.9)	15.6, 31.1	0.32
No	280 (24.5)	18.5, 31.5	192 (70.2)	57.7, 80.2	88 (29.8)	19.7, 42.3	
Percentage of children with at least three books (%)							
≥3	141 (26.1)	18.6, 35.2	116 (72.5)	58.6, 83.0	25 (27.5)	17.0, 41.4	0.60
<3	612 (73.9)	64.8, 81.3	453 (76.5)	66.9, 84.0	159 (23.5)	16.0, 33.1	
**Percentage of children with two or more toy types** (%)							
** ≥2**	**738 (99.4)**	**98.6, 99.7**	**562 (75.6)**	**67.9, 81.9**	**176 (24.4)**	**18.2, 32.0**	**0.02**
** <2**	**15 (0.6)**	**0.3, 1.3**	**7 (47.2)**	**23.9, 71.7**	**8 (52.8)**	**28.3, 76.1**	
**Children disciplined with violent methods** **(%)**							
** Yes**	**452 (59.9)**	**50.8, 68.4**	**319 (66.1)**	**55.7, 75.1**	**133 (33.9)**	**24.9, 44.3**	**< 0.001**
** No**	**301 (40.1)**	**31.6, 49.2**	**250 (89.5)**	**83.3, 93.5**	**51 (10.5)**	**6.5, 16.7**	

^a^ Stunting condition: no stunting (>−2 standard deviation height-for-age); stunting (≤−2 standard deviation height-for-age); ^b^ Missing value = 55; ^c^ Missing value = 6; ^d^ Chi-square.

**Table 3 ijerph-20-05174-t003:** Association between blood lead level and language z-score, stratified by stunting condition.

	Not Stratified Analysis		Stratified Analysis			
	Complete Sample		Stunted Children		Not Stunted Children	
	*n* Sample = 1394 *n* = 2,415,000		*n* Sample = 256 *n* = 437,500		*n* Sample = 1138 *n* = 1,945,000	
	β	95% CI	β	95% CI	β	95% CI
Child characteristics						
**Blood lead level** Non-detectable (< 3.3 μg/dL) **Detectable (≥ 3.3 μg/dL)**	−0.08	−0.53, 0.36	−0.40	−0.71, −0.10	−0.15	−0.36, 0.06
Stunting condition ^a^ Stunting No stunting	−0.16	−0.50, 0.18				
Blood lead level * Stunting condition Detectable vs. No stunting	−0.05	−0.55, 0.44				
**Sex** **Girl** **Boy**	0.07	−0.15, 0.29	−0.40	−0.67, −0.13	0.10	−0.13, 0.34
Maternal characteristics						
Maternal schooling ^b^ Middle school or higher Elementary	−0.05	−0.26, 0.16	−0.01	−0.35, 0.35	−0.05	−0.29, 0.18
Household characteristics						
SES ^c^ Lower Middle Higher	−0.06 −0.11	−0.47, 0.34 −0.43, 0.20	−0.47 −0.46	−1.10, 0.14 −1.00, 0.07	0.13 0.10	−0.20, 0.48 −0.21, 0.42
**Locality type** **Urban** **Rural**	−0.19	−0.41, 0.03	0.94	0.64, 1.23	−0.07	−0.31, 0.17
Country region North Center South	−0.15 −0.30	−0.51, 0.20 −0.66, 0.05	0.04 −0.23	−0.57, 0.66 −0.71, 0.24	−0.17 −0.33	−0.64, 0.28 −0.81, 0.14
Family care characteristics						
Preschool education Yes No	−0.10	−0.28, 0.07	0.03	−0.28, 0.34	−0.13	−0.33, 0.06
**Percentage of children with learning support** **Yes** **No**	−0.18	−0.37, 0.01	−0.26	−0.58, 0.06	−0.23	−0.46, −0.005
**Percentage of children with at least three books** **≥3** **<3**	−0.30	−0.58, −0.02	−0.57	−1.28, 0.12	−0.26	−0.55, 0.03
Percentage of children with two or more toy types ≥2 <2	−0.12	−0.73, 0.48	−0.31	−0.91, 0.29	−0.05	−0.73, 0.62
Children disciplined with violent methods Yes No	−0.10	−0.30, 0.10	0.14	−0.13, 0.42	−0.17	−0.38, 0.04

^a^ Stunting condition: no stunting (>−2 standard deviation height-for-age); stunting (≤−2 standard deviation height-for-age); ^b^ Missing value = 83; ^c^ Missing value = 6.

**Table 4 ijerph-20-05174-t004:** Association between blood lead level and not being developmentally on-track according to ECDI.

	*n* Sample = 753 *n* = 1,300,000		
	OR	95% IC	*p*-Value
Child characteristics			
Blood lead level Non-detectable (< 3.3 μg/dL) Detectable (≥ 3.3 ug/dL)	1.00 1.09	0.57, 2.07	0.783
Stunting condition ^a^ Stunting No stunting	1.00 0.76	0.30, 1.89	0.563
**Age** (**months**)	**0.91**	**0.86, 0.96**	**0.003**
Sex Girl Boy	1.00 1.02	0.51, 2.01	0.954
Maternal characteristics			
Maternal schooling ^b^ Middle school or higher Elementary	1.00 0.79	0.38, 1.64	0.532
Household characteristics			
**SES ^c^** **Lower** **Middle** **Higher**	**1.00** **4.28** **12.7**	**2.27, 8.05** **6.21, 25.7**	**<0.001** **<0.001**
Locality type Urban Rural	1.00 0.68	0.34, 1.36	0.276
**Country region** **North** **Center** **South**	**1.00** **2.73** **3.24**	**1.09, 6.80** **1.32, 7.94**	**0.031** **0.011**
Family care characteristics			
Preschool education Yes No	1.00 1.03	0.46, 2.34	0.925
Percentage of children with learning support Yes No	1.00 0.91	0.43, 1.95	0.827
Percentage of children with at least three books ≥3 <3	1.00 0.60	0.29, 1.23	0.165
Percentage of children with two or more toy types ≥2 <2	1.00 1.96	0.49, 7.86	0.336
**Children disciplined with violent methods** **Yes** **No**	**1.00** **0.13**	**0.06, 0.28**	**< 0.001**

ECDI: Early Childhood Development Index; Not on-track vs. on-track; ^a^ Stunting condition: no stunting (>−2 standard deviation height-for-age); stunting (≤−2 standard deviation height-for-age); ^b^ Missing value = 55; ^c^ Missing value = 6.

## Data Availability

Data are available on this website: https://ensanut.insp.mx/encuestas/ensanut100k2018/descargas.php, accessed on 1 July 2020.

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
