# Peer review of "Lead Exposure Can Affect Early Childhood Development and Could Be Aggravated by Stunted Growth: Perspectives from Mexico"

_ijerph, 2023, doi:10.3390/ijerph20065174_

Round 1
Reviewer 1 Report
1. I would recommend conducting proofreading and editing the entire manuscript. This will clean up majority of simple errors in spelling, punctuation, or grammar found throughout the manuscript, as well as correcting logical flow.
2. In Results section, it might be more helpful to add a data table for values by region (language z-scores, prevalence of children with detectable BPb, etc.). The manuscript makes a valid discussion on the need for regional approach in public health actions (lines 63-74). I think this is a very important implication of the study and adding a data table showing differences by region will strengthen that argument. Authors might want to consider noting this point in the Conclusion section, if they choose.
3. Reference to Table 3 is missing from Results section (page 10. Lines 8-16)
4. For some of the non-English works cited (about 16 of them), English translation of the titles were missing. It made it harder to check if the references are relevant to the research.
Reviewer 2 Report
See PDF Attached.

Reviewer 3 Report
Overall the study detailed in this paper is an important endeavor especially given the underrepresented nature of the population that is being studied.
While there is merit in this study, the English grammar and spelling needs to be fixed throughout the paper. In the first few lines, I note some errors:
Line 4-"Mayor" should be "major"
Line 5- "stunting" should be "stunted"
Line 10- the best grammar for this sentence is "Children with undernutrition are more vulnerable to lead exposure...."
There are many other errors that need to be fixed.
Besides, grammar and spelling, the authors should mention that iron deficiency is the primary micronutrient deficiency that increases vulnerability to lead exposure. since the authors mention that the data that they assessed contains nutritional information, relevant data seem to be missing from their analyses. Primarily hematological data (PCV, Hb, serum ferritin?)
By adding some markers of nutritional status, the paper will go from average to high impact in the opinion of this reviewer.
Round 2
Reviewer 2 Report
MS: ijerph-2239804
Title: Lead exposure can affect early childhood development and could be aggravated by stunted growth: Perspectives from Mexico
Comments for authors
I commend the authors on their responsivity to reviewers' comments and overall, they made a great effort to address my concerns. Hence, recommended for publication.